# Rac1-mediated membrane raft localization of PI3K/p110β is required for its activation by GPCRs or PTEN loss

Onur Cizmecioglu[1,2], Jing Ni[1,2], Shaozhen Xie[1,2], Jean J Zhao[1,2], Thomas M Roberts[1,2]*

[1]Department of Cancer Biology, Dana-Farber Cancer Institute, Boston, United States; [2]Department of Biological Chemistry and Molecular Pharmacology, Harvard Medical School, Boston, United States

**Abstract** We aimed to understand how spatial compartmentalization in the plasma membrane might contribute to the functions of the ubiquitous class IA phosphoinositide 3-kinase (PI3K) isoforms, p110α and p110β. We found that p110β localizes to membrane rafts in a Rac1-dependent manner. This localization potentiates Akt activation by G-protein-coupled receptors (GPCRs). Thus genetic targeting of a Rac1 binding-deficient allele of p110β to rafts alleviated the requirement for p110β-Rac1 association for GPCR signaling, cell growth and migration. In contrast, p110α, which does not play a physiological role in GPCR signaling, is found to reside in nonraft regions of the plasma membrane. Raft targeting of p110α allowed its EGFR-mediated activation by GPCRs. Notably, p110β dependent, PTEN null tumor cells critically rely upon raft-associated PI3K activity. Collectively, our findings provide a mechanistic account of how membrane raft localization regulates differential activation of distinct PI3K isoforms and offer insight into why PTEN-deficient cancers depend on p110β.

*For correspondence:
thomas_roberts@dfci.harvard.edu

## Introduction

The phosphoinositide 3-kinase (PI3K) pathway has been widely studied as a major regulator of cellular proliferation, metabolism and migration (*Fruman et al., 1998*; *Engelman et al., 2006*; *Cain and Ridley, 2009*). PI3Ks catalyze phosphorylation of the 3'OH group on phosphoinositides at the plasma membrane, which induces translocation of the Ser/Thr kinase Akt. Once recruited to the plasma membrane, Akt becomes activated by PDK1 and mTOR2 dependent phosphorylation events (*Alessi et al., 1997*; *Sarbassov et al., 2005*). Activation of PI3K signaling frequently occurs in human cancers (*Vivanco and Sawyers, 2002*).

The PI3Ks are organized into three classes based on their structural similarities and mechanism of activation. The most extensively studied class I PI3Ks are activated by growth factor receptors. Of these, class IA enzymes are activated via receptor tyrosine kinases (RTKs), small G-proteins and G-protein-coupled receptors (GPCRs) and PI3K signaling is functionally antagonized by the $PIP_3$ degrading PI3 phosphatase PTEN (phosphatase and tensin homolog) (*Sang et al., 2012*). p110α and p110β are the catalytic class IA isoforms with a ubiquitous expression pattern (*Jia et al., 2009*). Heterodimerization with the regulatory p85 subunit stabilizes p110 isoforms but at the same time keeps their kinase activity in check through inhibitory interactions (*Yu et al., 1998*; *Zhang et al., 2011*). Although both p110α and p110β form heterodimers with p85 and catalyze the same biochemical reaction upon activation, studies have indicated that they have distinct functions (*Foukas et al., 2006*; *Zhao et al., 2006*; *Jackson et al., 2005*; *Jia et al., 2008*; *Guillermet-Guibert et al., 2008*; *Ni et al., 2012*). p110α appears to mediate bulk of PI3K signaling downstream of RTKs whereas

p110β preferentially signals downstream of GPCRs. Notably p110α and β isoforms are involved in mutually exclusive signaling complexes with distinct members of small GTPase protein superfamily, Ras and Rac1 respectively (*Gupta et al., 2007*; *Fritsch et al., 2013*). However, we are far from a complete understanding of the molecular mechanisms that account for the full spectrum of their functional differences.

In cancer, p110α is the main isoform required for transformation by oncogenes (*Zhao et al., 2006*) and the encoding *PIK3CA* is frequently found to have suffered from one of two common activating hotspot mutations in many tumor types (*Samuels et al., 2004*). This reliance of tumors on p110α may be partially explained by the fact that it has much higher specific activity compared to p110β (*Zhao et al., 2006*; *Beeton et al., 2000*; *Knight et al., 2006*). Indeed in a GEM model of breast cancer driven by Her2, knockout of p110α blocked tumor formation while knockout p110β actually caused tumors to develop more quickly- a surprising result explained by a competition model where the less active p110β binds to the same limiting number of p85/p110 binding sites on Her2 and thus actually lowers signaling output compared to the case where only p110α is expressed and hence can bind to all of the PI3K binding sites on the receptor (*Utermark et al., 2012*). Notably tumors that have lost expression of PTEN are largely dependent on p110β (*Jia et al., 2008*; *Ni et al., 2012*; *Wee et al., 2008*), an unexpected outcome given its weaker kinase activity. While it is possible that some loss of PTEN tumors feature activation of GPCRs thus explaining their p110β dependence (*Rodriguez et al., 2016*), that does not appear to be a unifying theme.

The plasma membrane is the site of activation for numerous signaling cascades, and its organizing principles contribute to the specificity and potency of these sequences of biochemical reactions (*Kusumi et al., 2012*). The lipid raft hypothesis proposes that the preferential association among sphingolipids, sterols, and specific proteins confers upon the lipid bilayer the potential for lateral segregation of key signaling molecules, dramatically transforming the cell membrane from a passive bystander to a dynamic entity facilitating signaling through subcompartmentalization (*Lingwood and Simons, 2010*). Sphingolipid and cholesterol enriched membrane rafts play a key signaling role in T cell activation via the T cell synapse (*Gaus et al., 2005*), B cell activation (*Gupta and DeFranco, 2007*), focal adhesions, cell migration (*Gaus et al., 2006*), membrane trafficking in polarized epithelial cells (*van Meer et al., 1987*) and hormone signaling (*Márquez et al., 2006*).

Fluorescence correlation spectroscopy data implicate involvement of raft nanodomains in recruitment of Akt to the cell membrane upon $PIP_3$ production (*Lasserre et al., 2008*), and FRET-based Akt activity reporters uncovered a preferential activation of Akt in membrane rafts (*Gao and Zhang, 2008*). On the other hand PTEN was shown to localize selectively to nonraft membrane microdomains in human embryonic kidney cells, possibly further restricting Akt activation in these regions (*Gao et al., 2011*). However, although membrane microdomain compartmentalization is believed to be a crucial mechanism for achieving signaling specificity, the role of spatial partitioning in class IA PI3K signaling has remained elusive.

To address these questions, we combined simultaneous knock-out of p110α and p110β with genetic targeting that enabled us to express unique class IA PI3K isoforms directed to specific plasma membrane microdomains. Using this strategy, we investigated Akt activation, cellular proliferation and migration in response to growth factors, when expression of particular class IA PI3K isoforms was directed to membrane rafts or nonraft regions of the plasma membrane. We found that raft targeting of either p110α or p110β potentiates GPCR mediated activation of Akt. In addition, we determined that p110β required Rac1 binding for raft localization and Gβγ association for activation downstream of GPCRs, whereas raft-targeted p110α was dependent on EGFR activity. Notably we found that PI3K signaling was also dependent on raft integrity in PTEN null cancer cells. Taken together, these results indicate that any class IA PI3K catalytic isoform, when targeted to GPCR signaling permissive membrane microdomains, could activate Akt and maintain downstream PI3K signaling through distinct mechanisms. Finally, our data has novel implications concerning the optimal methods for inhibiting Akt activation in PTEN null tumors.

## Results

### Generation of Isogenic MEFs expressing p110α or p110β

To facilitate the study of the individual p110 isoforms, we first generated immortalized $p110\alpha^{flox/flox}$; $p110\beta^{flox/flox}$ mouse embryonic fibroblasts (MEFs). Expression of endogenous p110α and p110β is almost entirely lost upon transduction of the cells with an adenovirus (AdCre) expressing the Cre-recombinase (*Figure 1A*). These double knock-out (DKO) MEFs show a reduction of phosphorylated Akt (p-Akt) (*Figure 1A*, compare lanes 1–5 with lanes 6–10), have a blunted response in promotion of p-Akt and p-S6 to a variety of growth signals (*Figure 1B*, compare lanes 2–5 with lanes 7–10) and essentially cease to proliferate (*Figure 1C*), which is consistent with the key role of the PI3K pathway in proliferation and signaling.

To generate isogenic MEFs expressing either p110α or p110β, we ectopically expressed comparable levels of wild-type (wt) HA-p110α or HA-p110β in DKO MEFs (henceforth DKO 'add-back' is used to denote these MEFs) (*Figure 1D*). Notably, ectopic expression of either p110 isoform could rescue PI3K signaling defects and growth inhibition in the DKO MEFs (*Figure 1—figure supplement 1A,B*). The resulting isogenic DKO add-back MEFs were highly sensitive to small molecule inhibitors targeting the unique isoform expressed, in growth factor signaling and proliferation assays (*Figure 1E* and *Figure 1—figure supplement 1A,B*). These cells have allowed us to compare the two commonly expressed p110 isoforms in much more detail than has previously been possible.

### Rac1 binding constitutes a raft localization signal for p110β

Using this robust molecular genetic system, we attempted to address how spatial compartmentalization might contribute to the specificity of the activation of p110β by GPCRs. To determine whether wt p110β becomes differentially localized to membrane microdomains upon GPCR activation, we first analyzed triton sensitive and resistant membrane fractions in p110α-wt and p110β-wt DKO add-back MEFs. Our fractionation experiments demonstrated that p110β resides in detergent resistant membranes in addition to soluble and detergent sensitive fractions where p110α is primarily detected (*Figure 2—figure supplement 1A*). A detergent-free fractionation based on density gradient centrifugation in HMECs confirmed the notion that there are distinct pools of p110β localized to raft and nonraft membrane microdomains (*Figure 2A*). Of note, Rac1, which is known to interact with p110β displayed a similar fractionation pattern. p110α in comparison, can only be found in nonraft membranes. In addition to MEFs and HMECs; we analyzed HEK293, DU145, PC3, MCF7 and BT549 cancer cell lines by examination of triton sensitive and resistant membrane fractions and detected p110β in raft as well as nonraft fractions (*Figure 2—figure supplement 1B*).

As a canonical class IA PI3K isoform, p110β functions as a GPCR effector and interacts with the Gβγ component of the tripartite G-protein (*Guillermet-Guibert et al., 2008*; *Dbouk et al., 2012*). Physical association of p110β with Rac1 is also required to potentiate signaling (*Fritsch et al., 2013*). To test whether Rac1 or Gβγ may be responsible for localization of p110β to membrane rafts, we introduced point mutations to p110β that abolish its binding to Rac1 or Gβγ (*Fritsch et al., 2013*; *Dbouk et al., 2012*) (*Figure 2B*). Next, we generated DKO add-back MEF lines that express p110β-wt, Gβγ (GBM) or Rac1 (RBM) binding mutants as their only class IA PI3K isoform (*Figure 2C*). p110β-GBM or RBM DKO add-backs displayed retarded proliferation in comparison to either wt or p110β-wt DKO add-back MEFs (*Figure 2D*) and were largely deficient in migration through Transwell inserts towards chemoattractants (*Figure 2E*). Starvation and serum or GPCR agonist stimulation experiments demonstrated that although p110β deficient in either Gβγ or Rac1 binding, instigated Akt phosphorylation upon serum stimulation, both were significantly impaired in elevating p-Akt, in response to two GPCR agonists; lysophosphatidic acid (LPA) and sphingosine 1-phosphate (S1P) (*Figure 2F*). Interestingly, fractionation of raft versus nonraft membrane microdomains revealed that only p110β-GBM but not p110β-RBM localizes to rafts suggesting that Rac1 binding to p110β might constitute a raft localization signal (*Figure 2G*). We next attempted to verify this in HMECs, where p110β could readily be detected in triton resistant membrane fractions. We knocked-down Rac1 expression using Rac1 specific siRNAs and analyzed triton sensitive and resistant membrane microdomains. Downregulation of Rac1 led to a displacement of p110β from membrane rafts whereas raft and nonraft membrane marker distributions were unchanged (*Figure 2H*).

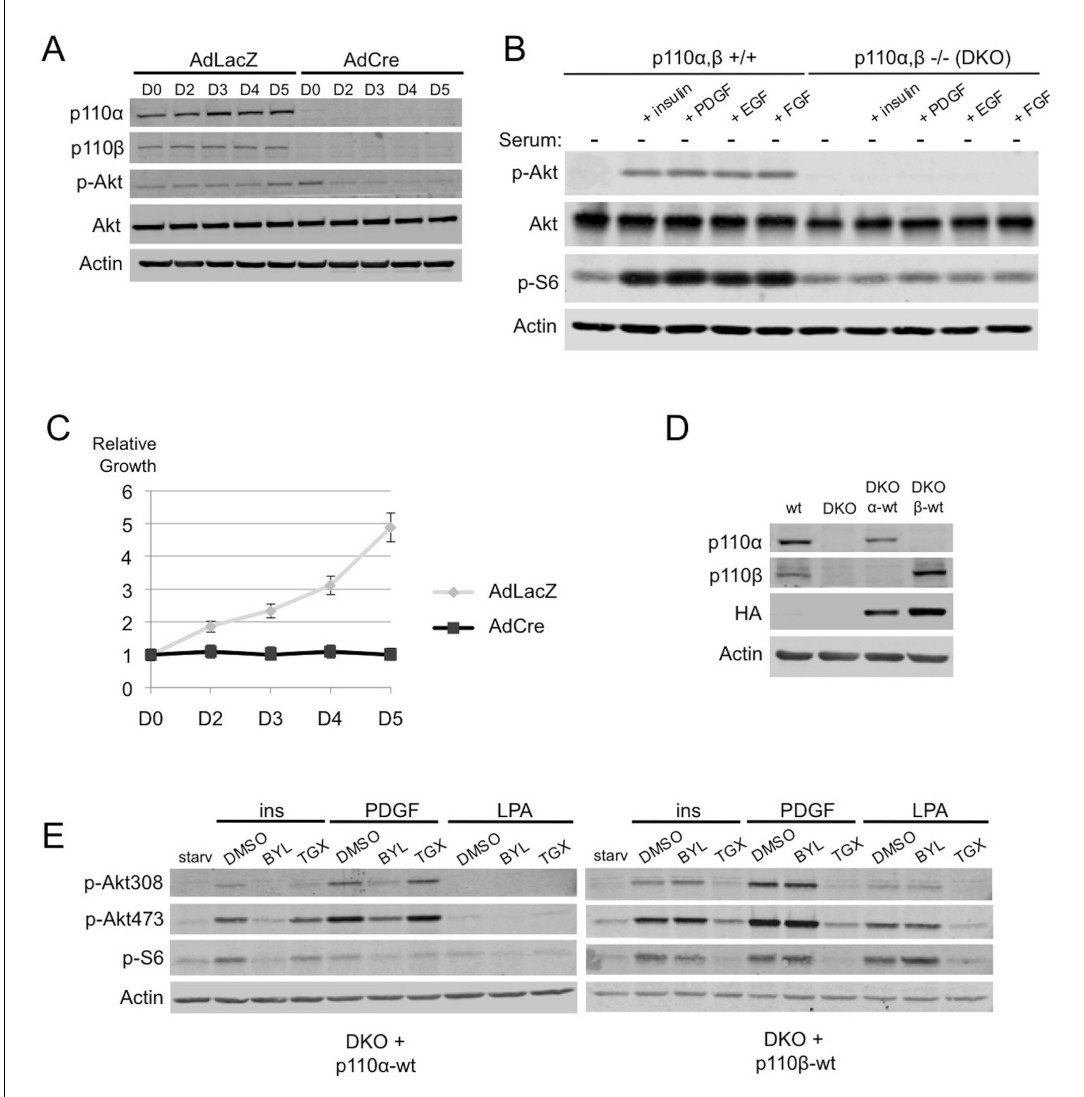

**Figure 1.** Simultaneous ablation of p110α and p110β disrupts PI3K pathway, which could be reconstituted by membrane microdomain targeting of either isoform in DKO MEFs. (**A**) $p110\alpha^{flox/flox}$; $p110\beta^{flox/flox}$ MEFs were treated with either AdLacZ or AdCre. 20 hr after the viral transductions, cells were harvested at indicated time points and immunoblots were performed. p-Akt (for T308) immunoblot depicts activation of PI3K pathway. (**B**) Wt or DKO MEFs were serum starved for 4 hr and stimulated with insulin, PDGF, EGF or FGF. Cells were then harvested and analyzed in immunoblots using the indicated antibodies (T308 for p-Akt and S235/236 for p-S6) to determine the extent of PI3K activation. (**C**) Proliferation kinetics of AdLacZ or AdCre treated MEFs were determined by crystal violet assays. Error bars depict standard deviation in 3 independent experiments. D denotes days. (**D**) Wt, DKO, DKO+p110α-wt and DKO+p110β-wt MEFs were analyzed in immunoblots for expression of p110α and p110β with the indicated antibodies. (**E**) DKO+p110α-wt and DKO+p110β-wt MEFs were starved and stimulated with insulin, PDGF and LPA in the presence of DMSO or isoform specific PI3K inhibitors, BYL-719 (BYL) and TGX-221 (TGX), specific for p110α and p110β, respectively. PI3K signaling efficiency in these cells was determined with p-Akt (for T308 and S473) and p-S6 (for S235/236) immunoblots.

The following figure supplement is available for figure 1:

**Figure supplement 1.** PI3K pathway could be reconstituted by unique expression of either p110α-wt or p110β-wt in DKO MEFs.

To determine whether raft targeting could compensate for Gβγ or Rac1 binding deficiencies in p110β, we constructed raft and nonraft-targeted p110β vectors using protein lipidation sequences derived from the Lyn tyrosine kinase and Kras respectively (*Gao and Zhang, 2008*) (*Figure 3A*) and generated p110β-Lyn and p110β-Ras DKO add-back MEFs. Analysis of homogenized samples fractionated via density gradient centrifugation confirmed enriched localization of p110β-Lyn to rafts

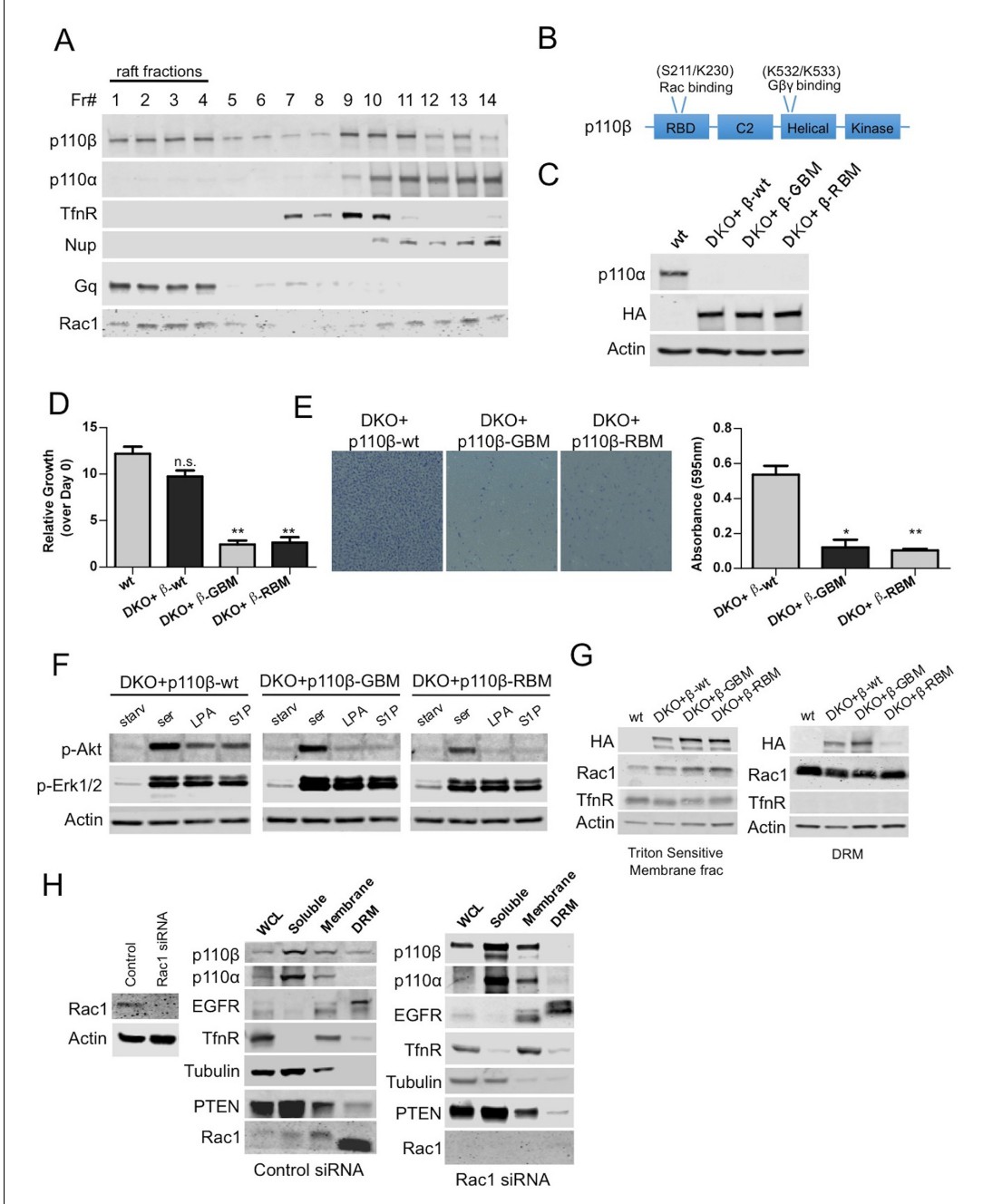

**Figure 2.** Rac1 binding domain constitutes a raft localization signal for p110β. (**A**) Detergent-free fractionation of HMECs on an Opti-prep gradient followed by western blots with the indicated antibodies. TfnR (transferrin receptor); marker for nonraft membranes. Nup (nucleoporin); marker for nuclear membranes. Gq; marker for membrane rafts. (**B**) Graphical depiction of the functional domains of p110β including the residues responsible for Rac1 and Gβγ interaction. (**C**) Wt, DKO+p110β-wt, DKO+p110β-Gβγ binding mutant (GBM) and DKO+p110β-Rac1 binding mutant (RBM) MEF samples were processed to determine levels of p110α and p110β expression. (**D**) The indicated cells were grown in 8% FBS-DMEM and cellular proliferation was assessed after a week. Error bars indicate standard deviation in 3 independent experiments. \*\*p<0.01, n.s. (not significant), p>0.05. (**E**) The indicated MEFs were analyzed for migration across transwell inserts; cells were detected with crystal violet staining. On the right, MEFs were quantified for migration through the transwell in 3 independent experiments with standard deviation. \*p<0.05, \*\*p<0.01. (**F**) The indicated MEFs were starved and stimulated with serum or GPCR agonists LPA and S1P. p-Akt (for T308) displays level of Akt activation and p-Erk1/2 depicts activation of MAPK pathway. (**G**) Indicated cells were fractionated into soluble, triton sensitive and triton resistant fractions. Triton soluble and resistant (DRM) fractions were analyzed in immunoblots; anti-HA antibodies were used to visualize the abundance of the p110β variants in those fractions. Anti-Rac1 antibody was used to demonstrate raft enrichment, whereas anti-TfnR immunoblot depicts enrichment of nonraft membranes. Anti-actin serves as loading control. (**H**) On the left, HMECs transfected with either control or Rac1 specific siRNAs were lysed and processed for western blot. On the right, siRNA

*Figure 2 continued on next page*

*Figure 2 continued*

treated cells were fractionated. WCL were analyzed to display overall levels of protein expression. Soluble, triton soluble (membrane) and triton resistant membrane fractions (DRM) were analyzed in immunoblots; Anti-Rac1 antibodies were used to assess level of Rac1 knock-down. Anti-EGFR antibodies were used as markers for DRM fractions, whereas anti-TfnR immunoblot depicts enrichment of nonraft membranes. Anti-tubulin immunoblot serves as a marker for soluble fractions.
The following figure supplement is available for figure 2:

**Figure supplement 1.** Membrane raft localization of p110β in different cell lines.

and p110β-Ras to nonraft membrane microdomains (*Figure 3B*). Elution of triton sensitive and triton resistant membrane fractions further demonstrated enrichment of the targeting plasmids at the desired microdomains (*Figure 3C*). p110β-Ras DKO add-back MEFs displayed a blunted response in Akt phosphorylation upon serum starvation and LPA stimulation whereas both p110β-wt and p110β-Lyn DKO add-backs had significant levels of p-Akt upon LPA stimulation (*Figure 3D*). Similarly, an increase in membrane-associated p-Akt was observed, when p110β-Lyn but not p110β-Ras DKO add-back MEFs were stimulated with LPA (*Figure 3—figure supplement 1*).

After confirming the involvement of raft-localized p110β in GPCR mediated activation of Akt, we added Lyn-domain to all versions of p110β (*Figure 4A*) and generated p110β-Lyn, p110β-Lyn GBM or p110β-Lyn RBM DKO add-back MEF lines (*Figure 4B*, left). As expected, add-back of p110β-wt enriched at rafts (*Figure 4B*, right) induced p-Akt in response to serum or LPA stimulation (*Figure 4C*, left). Interestingly, raft targeting restored p-Akt in p110β-Lyn RBM, but not in p110β-Lyn GBM DKO add-back MEFs upon LPA stimulation (*Figure 4C*, right). Taken together, these results strongly suggest that p110β-Rac1 association regulates raft recruitment of p110β, and imply that the Gβγ interaction with p110β might be independently required for its activation.

While p110β-Lyn is enriched in rafts, it is still moderately localized in nonraft regions, possibly through p85 SH2/SH3 domain-mediated molecular interactions. Therefore we tested whether raft localization is critical for compensating Rac1-binding deficiency in p110β-Lyn RBM. To this end, we selectively disrupted the formation of membrane rafts with the cholesterol-depleting agent, methyl-β-cyclodextrin (MβCD) in starvation-LPA stimulation experiments. Both p110β-Lyn and p110β-Lyn RBM failed to induce p-Akt when rafts were disrupted by MβCD, a defect which was rescued upon addition of excess cholesterol to the cells (*Figure 4D*). Moreover, the raft-excluded p110β-Ras allele was relatively insensitive to cholesterol depletion upon stimulation with either serum or PDGF, and promoted Akt phosphorylation in the presence of MβCD (*Figure 4E*). Notably LPA stimulation failed to induce p-Akt in either condition, presumably because p110β-Ras is excluded from rafts.

## Gβγ interaction is critical for p110β activation downstream of GPCRs

To identify the phenotypic consequences of raft-targeted, Gβγ or Rac1 binding deficient p110β expression, we first determined the growth rate of our p110β DKO add-back MEF lines under limiting amounts of mitogenic stimuli. Deficiency in Gβγ binding, significantly reduced rates of proliferation while p110β-Lyn RBM supported growth comparable to p110β-Lyn (*Figure 5A*).

Next, we investigated the function of raft-targeted p110β in cellular migration, which is known to be regulated by GPCR and Rac1 dependent signaling (*Dorsam and Gutkind, 2007*; *Bid et al., 2013*). Specifically we analyzed the efficiency of our DKO add-back MEFs in wound healing assays. Wound closure was monitored up to 12 hr upon scratching. p110β-Lyn RBM DKO add-back MEFs migrate more efficiently into the wound than p110β-Lyn GBM DKO add-back cells (*Figure 5B,C*). This observation is consistent with a distinct activating role of Gβγ on p110β, which cannot be compensated by relocalization of the molecule to rafts. Transwell migration assays using the same add-back lines reinforced the notion that selective raft targeting is compensatory for p110β-RBM whereas p110β-Lyn GBM DKO add-backs are defective in migration (*Figure 5D,E*). Taken together, this data is again consistent with the idea that the primary function of p110β-Rac1 binding lies in the placement of p110β in membrane rafts and thus creating a GPCR signaling competent PI3K module.

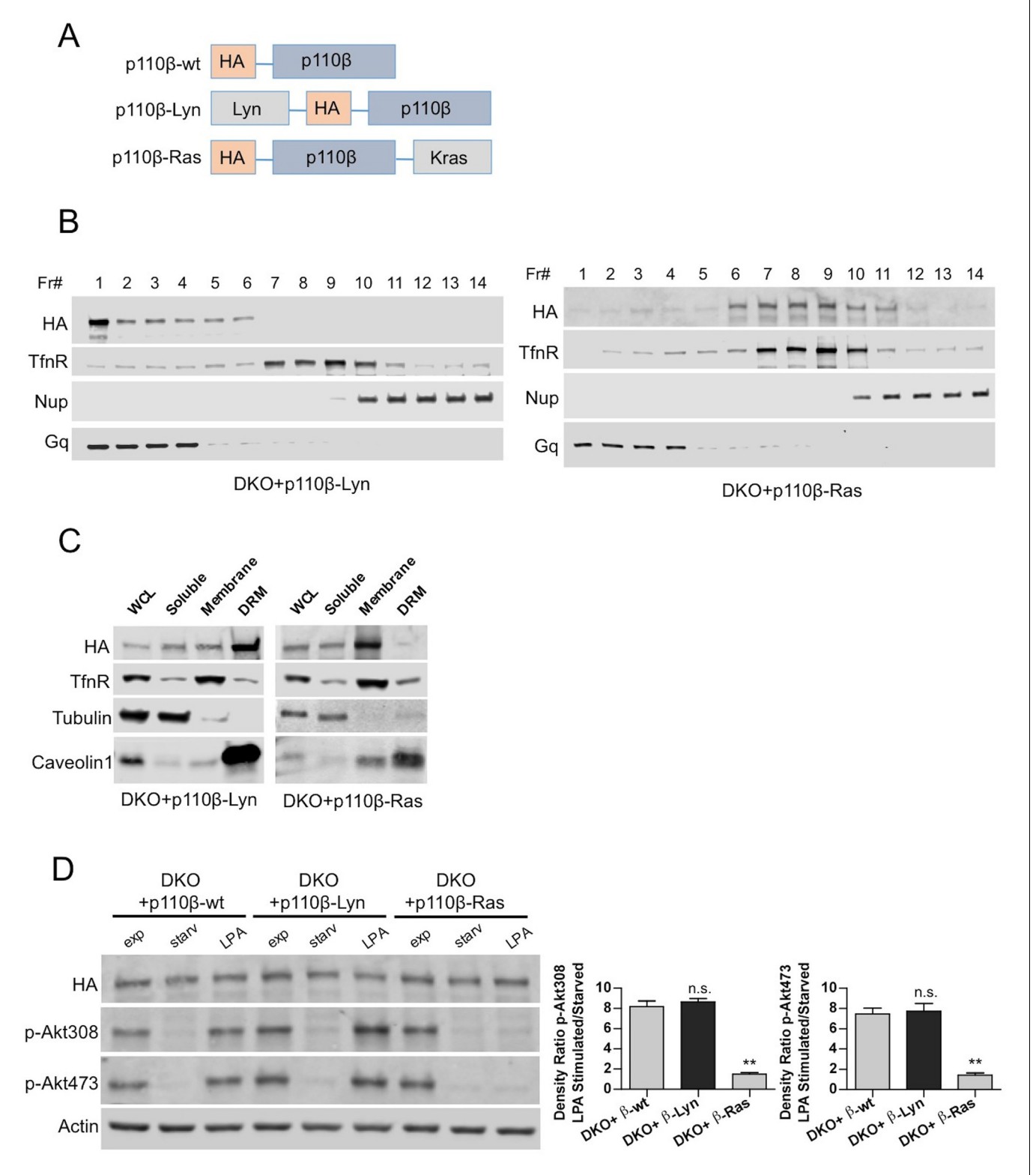

**Figure 3.** Raft-excluded p110β fails to induce Akt phosphorylation upon GPCR stimulation. (A) Schematic demonstration of p110β membrane microdomain targeting vectors. (B) Detergent-free fractionation of DKO+p110β-Lyn and DKO+p110β-Ras MEFs on an Opti-prep gradient followed by western blots with the indicated antibodies. TfnR; a marker for nonraft plasma membrane. Nup; a marker for nuclear membranes. Gq; a marker for membrane rafts. (C) The indicated MEFs were lysed and fractionated. WCL were analyzed to display overall levels of protein expression. Soluble, triton soluble (membrane) and resistant membrane fractions (DRM) were analyzed in immunoblots; anti-Caveolin1 antibodies were used as marker for DRM

*Figure 3 continued on next page*

*Figure 3 continued*

fractions. Anti-tubulin immunoblot serves as a marker for soluble fractions. (D) The indicated add-back MEFs were starved and stimulated with LPA. Anti-HA immunoblot demonstrates levels of exogenous p110β expression whereas anti-p-Akt antibodies (for T308 and S473) mark the activation state of Akt. Anti-actin antibodies were used as loading control. On the right, normalized anti-p-Akt T308 and S473 band intensity quantifications of the samples (mean of 3 independent experiments with standard deviation). Density ratio is the fold change between normalized band intensities of p-Akt signals in LPA stimulated over starved samples. **p<0.01, n.s. p>0.05.

The following figure supplement is available for figure 3:

**Figure supplement 1.** Raft-excluded p110β fails to induce activatory Akt phosphorylation upon GPCR stimulation.

## Raft Targeted p110α facilitates GPCR Signaling and has redundant functions with p110β

To test our hypothesis that Rac1 binding localized p110β to GPCR rich membrane microdomains where it could be further activated by a second signal, we attempted to see if we could render p110α, which is not normally GPCR responsive, sensitive to GPCR signaling by changing its membrane localization. To this end, we generated vectors that selectively target wt p110α to subdomains of the plasma membrane. Following our previous work, p110α was targeted to membrane rafts by using the Lyn-derived targeting motif and to nonraft membranes by the motif derived from Kras (*Figure 6A*). Analysis of homogenized samples fractionated via density gradient centrifugation confirmed enriched localization of p110α-Lyn to rafts, whereas p110α-Ras was more prominently localized to nonraft regions of the plasma membrane (*Figure 6B*). Elution of triton sensitive and triton resistant membrane fractions further demonstrated enrichment of the targeting plasmids at the desired microdomains (*Figure 6—figure supplement 1A*). Surprisingly, use of LPA led to a significant phosphorylation of Akt in cells expressing raft-targeted p110α-Lyn in a manner comparable to that seen in p110β-wt DKO add-back MEFs (*Figure 6C*). Activation of Akt was observed to a much lesser extent in cells expressing the nonraft membrane targeting p110α-Ras allele. Similarly, an increase in membrane-associated p-Akt was observed, when p110α-Lyn but not p110α-Ras DKO add-back MEFs were stimulated with LPA (*Figure 6—figure supplement 1B*).

We also examined p110 isoform localization in the PTEN null PC3 prostate cancer cell line that is dependent on p110β function for Akt activation and growth (*Ni et al., 2012*; *Hill et al., 2010*). In these cells, we knocked down p110β expression using p110β-specific shRNAs. Downregulation of p110β reduced p-Akt and p-S6 levels (*Figure 6—figure supplement 2A*). We then tested if expression of membrane targeting p110α alleles could restore Akt activation and downstream S6 phosphorylation. p110α-Lyn but not p110α-Ras expression compensated for p110β loss and restored p-Akt and p-S6 in p110β knockdown cells (*Figure 6—figure supplement 2B*, compare lanes 7 with 8), suggesting that raft-targeted p110α might have redundant functions with wt p110β.

## EGFR activity is critical for signaling mediated by raft targeted-p110α

Because raft localization of p110α is sufficient for activation of Akt in LPA signaling, we next focused on the mechanism by which p110α is activated downstream of GPCRs. Previous reports suggest that GPCR activation can also trigger widespread downstream signaling via modulation of various growth factor receptor pathways (*Hsieh and Conti, 2005*). For instance, LPA stimulation promotes metalloprotease-mediated transactivation of EGFR, and EGFR activity itself is responsible for major growth-promoting effects of GPCR signaling (*Prenzel et al., 1999*). Since p110α has no Gβγ or Rac1 binding site, we hypothesized that GPCR mediated EGFR activity might be critical for raft-localized p110α to modulate Akt function in LPA signaling. To test this idea, we selectively inhibited Gβγ, EGFR or PDGFR with gallein, lapatinib and crenolanib respectively, in serum starved and LPA stimulated p110α-Lyn or p110β-wt DKO add-back MEFs. EGFR activation as well as Akt and the downstream S6 phosphorylations were observed for both MEF lines upon stimulation with LPA (*Figure 6D*). In support of our hypothesis, EGFR but not Gβγ or PDGFR inhibition reduced p-Akt/S6 in p110α-Lyn DKO add-back MEFs (*Figure 6D*, compare lanes 3, 4 and 5). In contrast, Gβγ activity was critical for p110β-wt mediated GPCR signaling while EGFR inhibition had only a modest effect (*Figure 6D*, compare lanes 8 and 9). Consistent with these results, p110α-Lyn DKO add-back MEFs were more

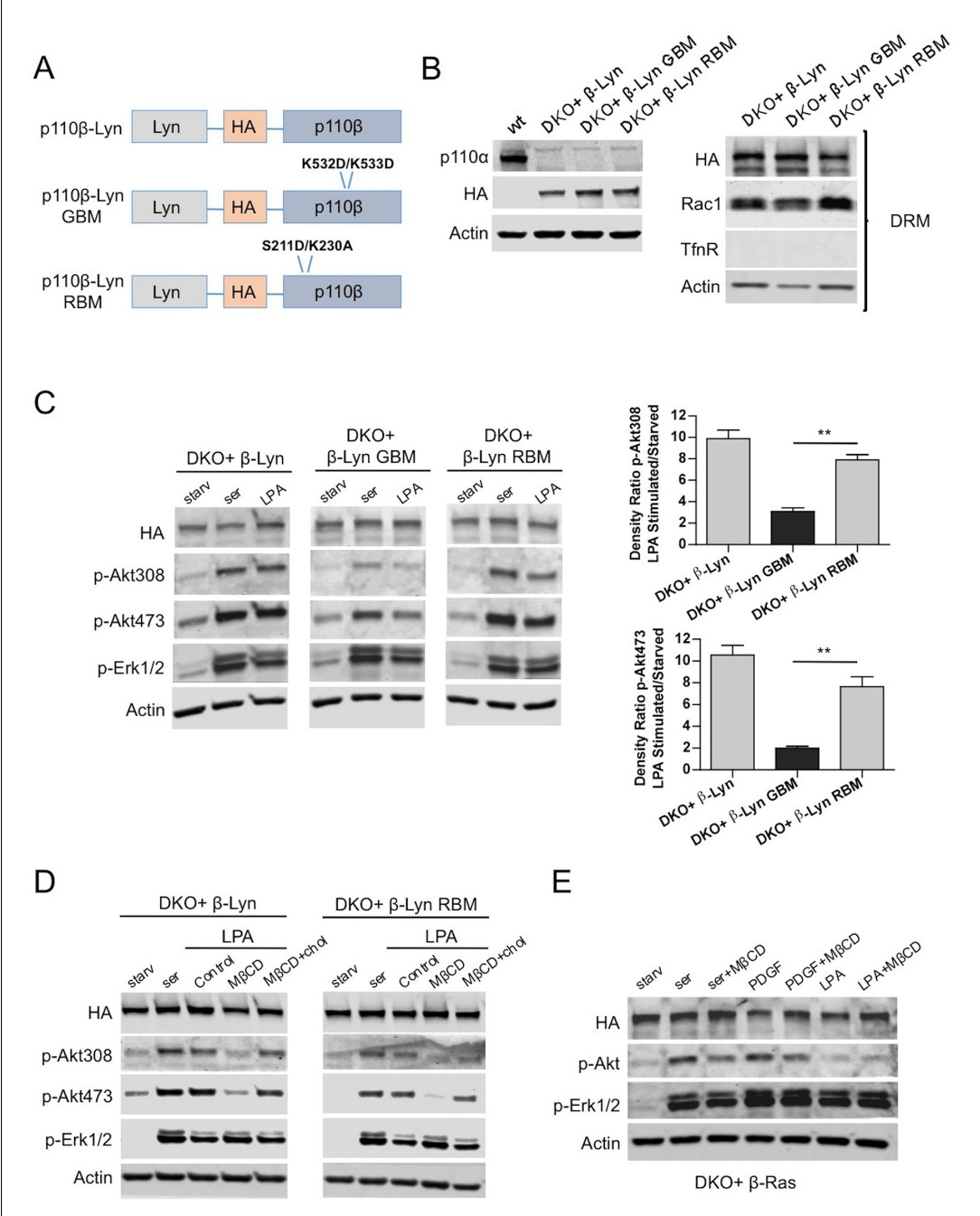

**Figure 4.** Raft targeting of Rac1-binding deficient p110β rescues Akt activation in GPCR signaling. (**A**) Schematic demonstration of p110β-Lyn domain membrane targeting vectors. (**B**) Lysates from the indicated MEFs were processed and analyzed for expression of p110α and β. On the right, DKO MEFs expressing the indicated p110β alleles were fractionated into soluble, triton sensitive and triton resistant fractions. Triton resistant fractions were analyzed in immunoblots; anti-HA antibodies were used to visualize the abundance of the p110β variants in those fractions. Anti-Rac1 antibody was used to demonstrate raft enrichment, whereas anti-TfnR immunoblot depicts contamination with nonraft membranes. Anti-actin immunoblot serves as loading control. (**C**) The indicated add-back MEFs were starved and stimulated with serum or LPA. Anti-p-Akt immunoblots on T308 and S473 display level of Akt activation and anti-p-Erk1/2 antibodies (for T202/Y204) depicts activation of MAPK pathway. On the right, density ratios of the normalized fold-increase in baseline Akt phosphorylation at T308 and S473 in starved vs. LPA stimulated states were quantified (mean of 3 independent experiments with standard deviation). **p<0.01. (**D**) The indicated MEFs were starved and stimulated with either serum or LPA in the presence of MβCD with or without addition of excess cholesterol. Anti-p-Akt immunoblots on T308 and S473 displays level of Akt activation and anti-p-Erk1/2 antibodies (for T202/Y204) depict activation of MAPK pathway (**E**) DKO+p110β-Ras add-back MEFs were starved and stimulated with serum, PDGF or
*Figure 4 continued on next page*

*Figure 4 continued*

LPA in the presence or absence of MβCD. Anti-p-Akt (for T308) and anti-p-Erk1/2 (for T202/Y204) immunoblots reveal degree of PI3K and MAPK activation respectively.

resistant to gallein in cellular proliferation assays (*Figure 6E*, left), whereas p110β-wt DKO add-backs were refractory to EGFR inhibition by lapatinib (*Figure 6E*, right). These results suggest that the EGFR activation by LPA stimulated GPCRs is required for raft-localized p110α to promote Akt phosphorylation and activation; whereas the canonical Gβγ pathway was essential for p110β-mediated GPCR signaling.

## Raft dependent PI3K function is essential for PTEN null cancer cells

Our results thus far suggest that raft integrity might play a more significant role for p110β dependent signaling. We and others have demonstrated that tumors deficient of phosphatase and tensin homolog (PTEN) are often dependent on the p110β isoform. Therefore we employed PTEN intact and PTEN null cancer cell line pairs and attempted to disrupt rafts by inhibiting cholesterol function. First, we treated PTEN positive DU145 and PTEN null PC3 prostate cancer cells with increasing doses of the cholesterol-depleting agent MβCD. PC3 cells were significantly more sensitive to MβCD treatment than DU145 cells (*Figure 7—figure supplement 1A,B*). Next we used PTEN wt MCF7 and PTEN null BT549 breast cancer cells to further test our hypothesis. Along the same line, BT549 breast cancer cells were more susceptible to MβCD than MCF7 cells (*Figure 7—figure supplement 1C,D*). As an independent approach, we utilized simvastatin, an FDA-approved inhibitor of the enzyme HMG-CoA reductase, which catalyzes the rate-limiting step (HMG-CoA to mavelonic acid) in cholesterol biogenesis in order to lower cholesterol levels. Once again, PC3 cells were considerably more sensitive to inhibition of cholesterol synthesis than DU145 cells. A simvastatin concentration of 1 μM was sufficient to potently inhibit proliferation of PC3 cells, which did not impede growth of DU145 cells (*Figure 7A,B*). Next we treated PTEN wt MCF7 and PTEN null BT549 cells with simvastatin. Treatment of BT549 cells with simvastatin led to a significant impediment of proliferation at 1–2 μM concentrations, which was not growth inhibitory for MCF7 cells (*Figure 7C,D*). Another PTEN wt breast cancer cell line, T47D was similarly refractory to growth inhibitory effects of simvastatin (*Figure 7—figure supplement 1E,F*). These results suggest that PTEN null

cancers might be more sensitive to inhibition of raft function which in turn negatively influences p110β dependent PI3K signaling. In an attempt to restore PI3K pathway activation, we expressed constitutively active, oncogenic mutants of p110α, H1047R and E545K in PC3 cells. Notably this lead to an activation of Akt as anticipated (*Figure 7E*). Next, we treated these cells with the cholesterol-lowering agent, simvastatin. Although control PC3 cells were quite susceptible to simvastatin treatment, PC3-p110α H1047R or E545K expressing cells proliferated more robustly in the presence of previously deleterious doses of the inhibitor (*Figure 7E and F*). A biochemical analysis of these cells showed that Akt activation was repressed in PC3 cells treated with simvastatin and this repression was relieved by p110α-H1047R or E545K expression (*Figure 7G*). These results suggest that raft specific inhibition of PI3K function in PC3 cells can be partially alleviated by expression of constitutively active oncogenic alleles of p110α, which presumably utilize raft independent modes of activation. As a complementary approach, we re-expressed either a wt or a catalytically dead version of human Pten (Pten C124S) in PC3 and BT549 cells. Restoration of wt Pten lead to a reduction in Akt activation as judged by levels of pAkt (*Figure 7—figure supplement 2A and D*). Interestingly, Pten expressing PC3 and BT549 cells became more resistant to increasing concentrations of simvastatin, while cells expressing Pten C124S were almost as sensitive to simvastatin as the parental cell lines (*Figure 7—figure supplement 2B,C,E and F*, E and F). These results are in support of the notion that loss of Pten in certain cancers might lead to a dependence on raft mediated PI3K signaling.

## Discussion

Signal transduction via the PI3K pathway requires a sequence of highly orchestrated molecular events transpiring at the plasma membrane. Nevertheless, it is still unclear how local signaling is modulated and how membrane microdomain architecture might influence the signal output. We

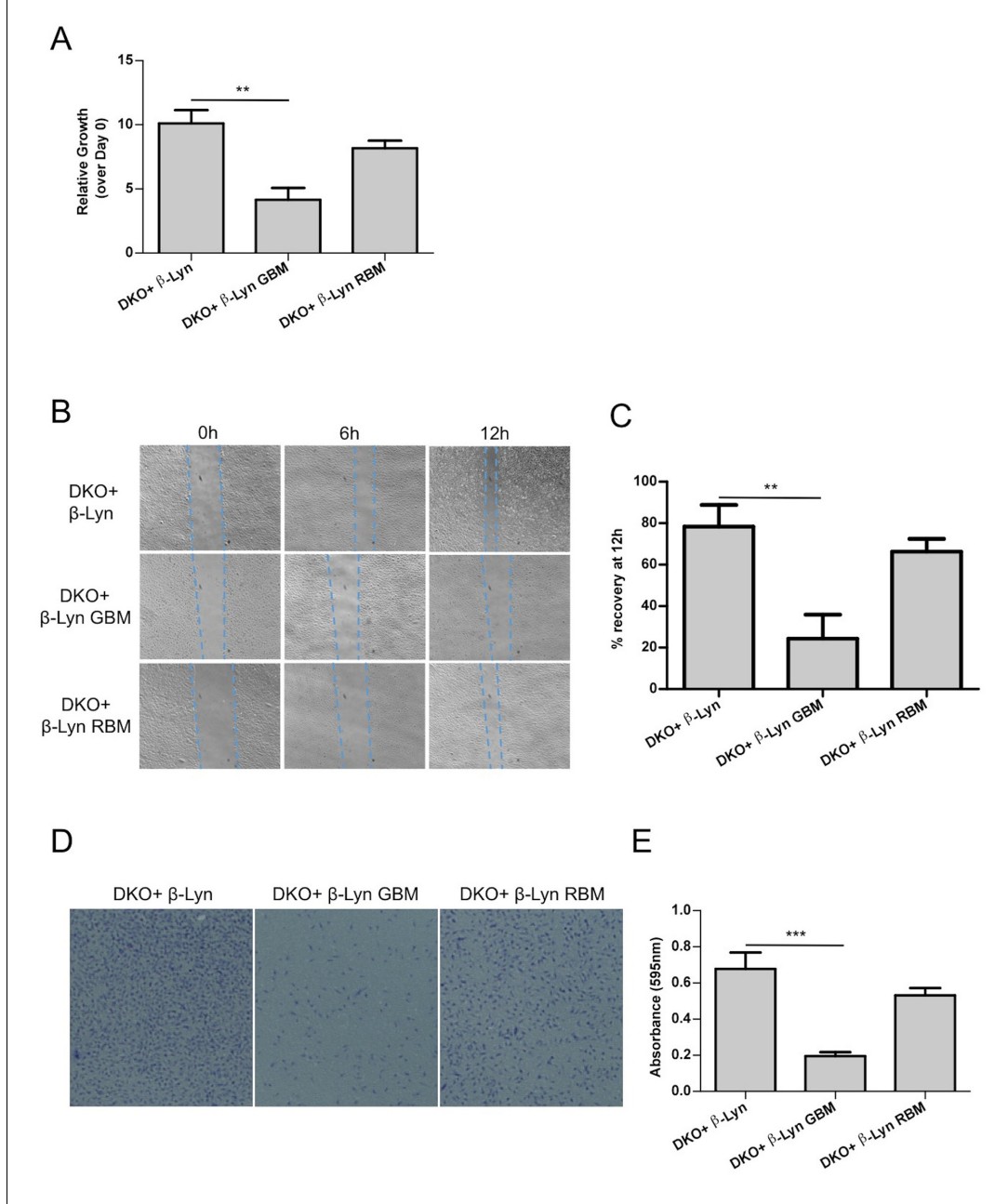

**Figure 5.** Defects in proliferation and motility are restored upon targeting Rac1-binding deficient p110β to membrane rafts whereas Gβγ interactionremains to be essential for p110β function. (**A**) Wt and the indicated DKO add-back MEFs were analyzed for cellular proliferation in crystal violet assays (mean of 3 independent experiments with standard deviation). **p<0.01. (**B**) Wound healing assays were performed on the indicated MEFs. Images of the cells were captured at the indicated time points; dashed lines represent leading edge of the cells moving into the inflicted wound. (**C**) Graph depicts the mean percentage of wound healing at the end of 12 hr for cells indicated in (**B**); in 3 independent experiments with standard deviation. **p<0.01. (**D**) DKO+p110β-Lyn, DKO+p110β-Lyn GBM and DKO+p110β-Lyn RBM add-back MEFs were analyzed for migration across transwell inserts; cells were detected with crystal violet staining. (**E**) Cells in (**D**) were quantified for migration through the transwell in 3 independent experiments, mean absorbance (at 595 nm) with standard deviation is depicted. ***p<0.001.

have used a molecular genetic system that allowed us to express specific membrane-targeted iso-forms of either p110α or p110β in an isogenic p110α/p110β DKO background. This genetic system enabled functional comparisons between wild type and membrane-targeted alleles of class IA PI3Ks. Interestingly for p110β, raft targeting alleviates the requirement for Rac1 binding. However, Gβγ

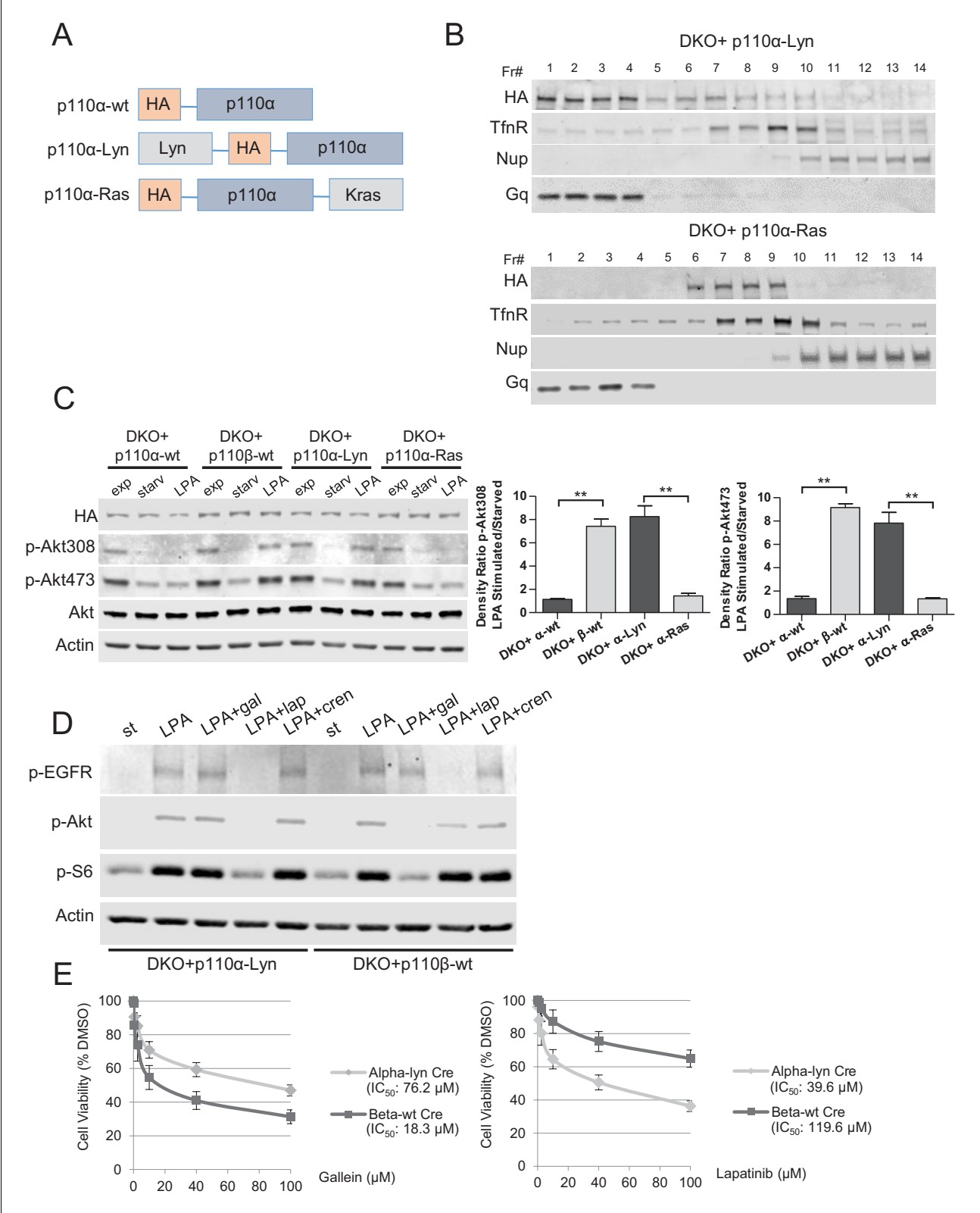

**Figure 6.** Raft-targeted p110α induces Akt phosphorylation upon GPCR signaling via EGFR activity. (**A**) Schematic demonstration of p110α membrane microdomain targeting vectors. (**B**) Detergent-free fractionation of DKO+p110α-Lyn and DKO+p110α-Ras MEFs on an Opti-prep gradient followed by western blots with the indicated antibodies. TfnR; a marker for nonraft plasma membrane. Nup; a marker for nuclear membranes. Gq; a marker for membrane rafts. (**C**) The indicated add-back MEFs were starved and stimulated with LPA. Anti-p-Akt antibodies (for T308 and S473) mark the activation

*Figure 6 continued on next page*

*Figure 6 continued*

state of Akt. Anti-Akt and anti-actin immunoblots were used as loading controls. On the right, normalized density ratios of the mean fold-increase in baseline Akt phosphorylation at T308 and S473 in starved vs. LPA stimulated states. Graphs denote mean of 3 independent experiments with standard deviation. **$p<0.01$. (D) DKO+p110α-Lyn and DKO+p110β-wt MEFs were starved and stimulated with LPA in the presence of small molecule inhibitors targeting Gβγ, EGFR or PDGFR. Anti-p-EGFR (for Y1068), anti-p-Akt (for T308) and anti-p-S6 (for S235/236) immunoblots depict activation of EGFR, Akt and downstream signaling. Gal denotes gallein, a Gβγ inhibitor; lap depicts lapatinib, an EGFR inhibitor and cren denotes crenolanib, a PDGFR inhibitor. (E) The indicated MEFs were treated with 0, 0.1, 1, 2.5, 10, 40 or 100 µM of lapatinib or gallein in proliferation assays. Cellular growth was assessed after five days in 2% FBS-DMEM. Error bars denote standard deviation in 3 independent experiments.

The following figure supplements are available for figure 6:

**Figure supplement 1.** Membrane targeting p110α vectors selectively enrich p110α in the desired microdomain.

**Figure supplement 2.** Raft-targeted p110α has redundant functions with wt p110β.

association is unable to be rescued by raft targeting. These results indicate that Rac1-p110β interaction may be necessary for localization of p110β to GPCR signaling permissive membrane microdomains, whereas its association with Gβγ remains crucial for the activation of its lipid kinase activity upon ligand binding to GPCRs. Interestingly, even p110α, which is usually unresponsive to GPCR activation, becomes responsive if properly localized. Finally, our data has implications concerning the optimal methods for inhibiting Akt activation in PTEN null cancers (*Figure 7H*).

Our DKO add-back MEF lines were extremely well behaved when tested by the guidance of existing literature: e.g. they were responsive to the relevant PI3K class IA isoform-specific small molecule inhibitors for both signaling and growth. Our efforts in enriching p110α and p110β in membrane rafts and nonraft regions using targeting sequences derived from the Lyn kinase and Kras were also clearly successful as judged by several means of membrane fractionation. Minimal off-target localization with these p110 constructs might be attributed to the indirect interactions of p85 mediated via its SH2 and/or SH3 domains.

The resulting targeting constructs were utilized to study the role(s) of microdomain compartmentalization in growth factor signaling. Serum starvation and stimulation experiments revealed that GPCR activation could be communicated to Akt when either p110α or p110β was adequately localized in membrane rafts. One fundamental difference between these PI3Ks however, is in their intrinsic membrane binding properties. p110β, by virtue of its association with Rac1, is naturally localized to membrane rafts. On the other hand, p110α does not reside in rafts under physiological conditions. However, when it is artificially targeted to the rafts, p110α is capable of functioning downstream of GPCRs. Moreover, raft-targeted p110α acts in a redundant manner with p110β in PTEN null PC3 cells.

While membrane raft localization is essential for p110 activation by GPCRs, whether it occurs via Rac binding for p110β in the physiological case or via a Lyn-tag for p110α, the second step in p110 activation by GPCRs can be more varied. In the physiological case p110β is activated via its interaction with Gβγ. However, we found that EGFR activity is essential for raft-localized p110α mediated Akt phosphorylation. EGFR is enriched in raft fractions, and its activity has been intimately linked to GPCR signaling. LPA stimulation, for instance, was reported to induce EGFR phosphorylation and transactivation (*Daub et al., 1997*). Membrane bound metalloproteases and non-receptor tyrosine kinases like Src and Pyk2 are implicated in this regulation (*Prenzel et al., 1999*; *Andreev et al., 2001*). Our results suggest that LPA stimulated-EGFR could contribute to activation of raft-localized p110α, whereas the canonical Gβγ association is the prominent form of activation for p110β. However, we cannot rule out a modest contribution of EGFR in raft-localized p110β activation, as we detected residual activation of Akt in cells expressing raft-targeted, Gβγ binding deficient p110β upon LPA treatment. Notably p110β may also be dependent on Rac mediated raft localization for its activation in the non-physiological conditions of PTEN loss. However, in the case of p110β activation in response to PTEN loss yet another mechanism may be involved as the second step in p110β activation- further experimentation will be required to test this point.

Our findings confirm the existing literature that both Rac1 and Gβγ binding are instrumental for p110 function downstream of GPCRs (*Fritsch et al., 2013*; *Dbouk et al., 2012*) and allow us to place

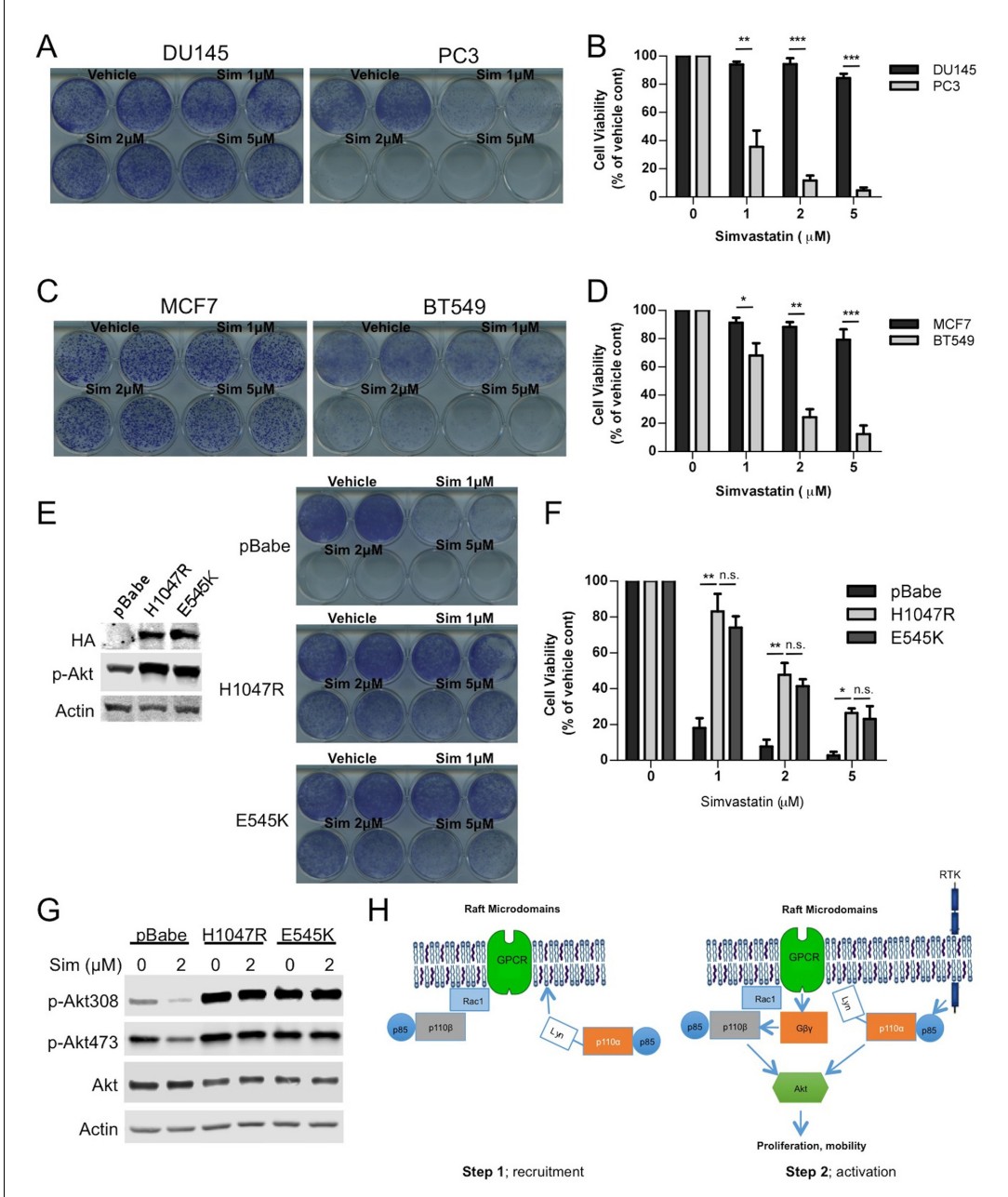

**Figure 7.** Raft dependent PI3K function is critical for PTEN null cancer cells. (**A**) DU145 and PC3 prostate cancer cell lines were seeded on 12-well plates and were treated with indicated doses of simvastatin for a week. Crystal violet assays determined the extent of cell growth. (**B**) Quantification of cellular proliferation assays in (**A**), graph displays mean of 3 independent experiments with standard deviation. $**p<0.01$, $***p<0.001$. (**C**) MCF7 and BT549 breast cancer cell lines were seeded on 12-well plates and were treated with indicated doses of simvastatin for a week. Crystal violet assays determined the extent of cellular growth. (**D**) Quantification of cellular proliferation assays in (**C**), graph depicts mean of 3 independent experiments with standard deviation. $*p<0.05$, $**p<0.01$, $***p<0.001$. (**E**) PC3 cells expressing empty control plasmid, p110α-H1047R and p110α-E545K vectors processed for western blot analysis with the indicated antibodies. (**F**) Indicated cells were treated with increasing doses of simvastatin for a week. Crystal violet assays determined the extent of cellular growth. (**G**) Quantification of cellular proliferation assays in (**F**), graph depicts mean of 3 independent experiments with standard deviation. $*p<0.05$, $**p<0.01$, n.s. $p>0.05$. (**H**) Biochemical analysis of PC3 cells in (**E**) treated with 2 μM simvastatin for 36 hr. Anti-p-Akt immunoblots on T308 and S473 display level of Akt activation. Anti-Akt and anti-actin antibodies were used as loading control. (**I**) Model representing PI3K signaling downstream of GPCRs in a two-step process. Rac1 mediates raft recruitment of p110β while p110α is elusive from rafts under physiological conditions. Once localized to GPCR signaling permissive membrane microdomains, either p110α or p110β can be activated via distinct mechanisms; p110α activity can be triggered via lateral activation of EGFR by GPCRs, whereas p110β can be activated via the canonical Gβγ pathway.

*Figure 7 continued on next page*

*Figure 7 continued*

The following figure supplements are available for figure 7:

**Figure supplement 1.** Critical dependency of PTEN null cancer cell lines to raft associated PI3K signaling.

**Figure supplement 2.** PTEN null cancer lines become sensitized to simvastatin upon re-expression of wild-type PTEN.

them in a two-step model for p110 activation. Raft targeting of mutant versions of p110 enabled us to infer that Rac1 binding, at least in a PTEN wild-type setting, is crucial for localization of p110β into signaling permissive membrane microdomains. Gβγ interaction on the other hand, provides the basis for p110β activation upon ligand-induced stimulation of GPCRs. We further verified the involvement of raft microdomains in LPA mediated GPCR signaling via cholesterol deprivation experiments. Cholesterol depletion specifically inhibited activating Akt phosphorylations whereas Erk phosphorylation was largely unaffected. These results indicate that activation of the MAPK signaling cascade by GPCRs is not crucially dependent on raft integrity, reinforcing the notion that critical role of membrane compartmentalization may be unique to PI3K/Akt signaling.

The higher abundance of PTEN in nonraft membrane regions seen here and in earlier work (*Gao et al., 2011*) might be an important determinant in localizing signaling as it may prevent unnecessary propagation of signaling cascades initiated in rafts to nonraft regions. Thus the reported selective localization of PTEN in nonraft domains suggests that it might exert relatively little control on raft recruitment of p110β and activation of Akt, while preventing the resulting signals from propagating outside raft domains. The elevated signaling potency observed when either p110α or β is targeted to rafts could be explained by a regional lack of PTEN dependent negative regulation. In tumor cells lacking PTEN, such p110β dependent signaling could spread to larger portions of the cells greatly strengthening PI3K signaling. However the fact that p110β signaling in PTEN null cells is still dependent on raft integrity suggests that the second, Rac-independent 'Gβγ like' signal activating p110β is likely to initiate in rafts in the absence of PTEN. We do not feel that this second signal is Gβγ itself in most PTEN null cells as Akt activation is often not blocked by GPCR antagonists.

Notably we have found that knock down of PTEN in our MEFs does not render their growth dependent on p110β or sensitive to raft disruption suggesting that while PTEN loss may be necessary for p110β activation (in rafts as demonstrated by the data in *Figure 7*) it is not sufficient. Similarly the Vanhaesebroeck group has found that not all tumors arising in mice with global knockout of a single PTEN allele are dependent on p110β (*Berenjeno et al., 2012*). Thus additional parameters seem to be involved in optimal activation of p110β in a PTEN null setting.

Use of simvastatin, an FDA approved cholesterol lowering drug, hampered proliferation of PTEN null cancer cells which might pinpoint a plausible 'addiction' of these cancer cells to raft mediated PI3K function. Based on these findings, a combinatorial treatment regimen employing PI3K inhibitors and cholesterol lowering statins might have the potential to treat a broad range of cancers with minimal undesirable side effects.

In summary, our findings establish a novel distinction between p110α and p110β, and offer new insight into the mechanistic basis of p110 activation by GPCR signaling. Here, we propose that recruitment and activation constitute two distinct steps in PI3K/Akt signaling downstream of GPCRs (*Figure 7H*). Once properly placed, either p110α or β molecules can be activated via alternative GPCR signaling cascades owing to the versatility of signaling components activated by a liganded GPCR. Our data warrant further work on the role of membrane partitioning in regulation of the PI3K/Akt pathway and offer novel therapeutic aspects concerning treatment of PTEN null cancers.

## Materials and methods

### Vector construction and shRNA sequences

Lyn and Kras tagged constructs (*Gao and Zhang, 2008*) were generated by in frame fusion of the N-terminal part of the Lyn kinase (GCIKKSKRKDKD, for myristoylation and palmitoylation) at the 5' end or the C-terminal part of Kras (KKKKKSKTKCVIM, CAAX motif for prenylation) at the 3'end of a

pBABE p110α or p110β vector with an N-terminal HA-tag respectively. p110β Gβγ (K532D/K533D) and Rac1 (S211D/K230A) binding mutants were generated using QuikChange site-directed mutagenesis kit (Agilent, Santa Clara, California) and sequence verified. The sequence of the shRNA targeting human p110β transcripts is as follows; CATTCAGCTGAACAGTAGCAA. shGFP sequence is GCAAGCTGACCCTGAAGTTCAT. pBabeL-Pten wt and pBabeL-Pten C124S plasmids were kind gifts from William Sellers (Addgene plasmids #10785 and #10931).

## Generation of immortalized mouse embryonic fibroblasts (MEFs), cell culture and generation of stable cell lines

HEK293, $p110\alpha^{flox/flox}$; $p110\beta^{flox/flox}$ MEFs and their derivatives were grown at 37°C in 5% $CO_2$ in Dulbecco's modified Eagle's medium (DMEM, including 4.5 g/L D-glucose, L-glutamine and 110 mg/L sodium pyruvate) supplemented with 8% fetal bovine serum (FBS, Gemini-Bio, West Sacramento, California) and penicillin, streptomycin (100 IU/ml and 100 µg/ml respectively, Gibco). Human mammary epithelial cells (HMECs) were generated and cultured as described (*Zhao et al., 2005*). DU145, PC3, MCF7 and BT549 cells were acquired from ATCC and were not further authenticated. They were cultured in RPMI medium (Gibco, Waltham, Massachusetts) supplemented with 8% FBS under standard conditions. All cell lines used were negative for mycoplasma contamination.

$p110\alpha^{flox/flox}$; $p110\beta^{flox/flox}$MEFs were prepared from embryos at embryonic day 13.5 post-fertilization. These primary MEFs were immortalized using the standard 3T3 protocol (*Meek et al., 1977*). Floxed MEFs were treated with AdCre (Iowa Viral Vector Core, Iowa City, Iowa) to generate knockout cells or with AdLacZ for control. Add-back MEF lines were generated by introducing the construct of interest into the cells first and then treating them with two rounds AdCre infection. For RNAi experiments, PC3 cells were treated with 1 µg/ml of doxycycline for 48 hr.

Control siRNAs (AM4629) and siRNAs targeting human Rac1 (ID 164723) were ordered from Thermo Fisher (Waltham, Massachusetts, transfected into HMECs at 50 nM concentration using lipofectamine 2000 (Invitrogen, Waltham, Massachusetts) according to the manufacturer's instructions. 48–72 hr post transfection; cells were harvested and subjected to immunoblot or fractionation assays.

Amphotropic retroviruses were produced by transfection of HEK293 cells with packaging plasmids encoding Vsv-g, gag-pol and a retroviral vector encoding the gene of interest using lipofectamine 2000 (Invitrogen) according to the manufacturer's instructions. For production of amphotropic lentiviruses, same cells were transfected with Vsv-g, Delta 8.9 and pLKO.tet on vector encoding an shRNA sequence of interest under standard instructions. Culture supernatants containing retro or lentivirus were collected twice at 48 hr and 72 hr posttransfection, pooled together, sterile filtered (Thermo Scientific) and then used immediately.

## Growth factor stimulation and drug treatment

Cells were starved for 4 hr and stimulated with insulin (2 µg/ml), EGF (20 ng/ml), LPA (20 µM, all Sigma), PDGF (20 ng/ml), FGF (10 ng/ml, both Miltenyi Biotech, Germany), or 5% FBS for 5 min. Cells were washed with and scraped into ice-cold PBS and then lysed in RIPA buffer (Westnet, Canton, Massachusetts) containing protease and phosphatase inhibitor cocktails (both Roche Diagnostics, Indianapolis, Indiana), 1 mM sodium orthovanadate (Cell Signaling, Danvers, Massachusetts) and 1 mM dithiothrietol (DTT, Bio-Rad, Hercules, California). p110α specific-inhibitor BYL-719 (1 µM), p110β specific-inhibitor TGX-221 (2 µM), pan-PI3K inhibitor GDC-0941 (1 µM), pan-Akt inhibitor MK-2206 (1 µM, all Selleck Chem, Houston, Texas) were used in proliferation experiments. In growth factor stimulation experiments BYL-719, TGX-221, Gβγ inhibitor Gallein (5 µM, Santa Cruz, Dallas, Texas), EGFR inhibitor Lapatinib (5 µM), PDGFR inhibitor Crenolanib (0.2 µM, both Selleck), MβCD (5 mM) and water soluble cholesterol (2 mM, both Sigma, St. Louis, Missouri) were added to serum starved cells 30 min before stimulation.

## Antibodies and western blotting

10–20 µg of total protein was separated with 7.5–12% SDS-PAGE and transferred to nitrocellulose membranes (Bio-Rad). The membranes were blocked with 5% non-fat milk (LabScientific, Highlands, New Jersey) in TBS for 30 min and then incubated with primary antibodies in TBS containing 5% milk and 0.1% Tween-20 overnight at 4°C. After rinsing with TBS-0.1% Tween, membranes were

incubated with the secondary antibodies diluted in 5% milk-TBS for 1.5 hr at RT. Bands were detected by using Odyssey CLx Scanner. Band intensity quantifications were performed using Image Studio 3.1. Normalizations were performed either relative to the actin or Akt band intensities. Western blots were performed with antibodies against p110α (#4249), p110β (#3011), Akt (#4691), phospho-Akt T308 (#13038), phospho-Akt S473 (#9271), phospho-Erk 1/2 T202/Y204 (#9101), phospho-S6 S235/236 (#2211), phospho-S6 S240/244 (#5364), phospho-EGFR Y1068 (#2236), S6 (#2217), HA (#2367), TfnR (transferrin receptor) (#13208), Nup (nucleoporin) (#2598), EGFR (#4267), PTEN (#9188) (all Cell Signaling), Rac1 (05–389) (Millipore, Billerica, Massachusetts), β-actin (A5441), α-tubulin (T5168) (Sigma), Caveolin1 (MA3-600) (Thermo Fisher) and Gq (sc-392) (Santa Cruz). IRDye 800CW Goat anti-mouse or anti-rabbit secondary antibodies (Li-Cor, Lincoln, Nebraska) were used.

## Membrane preparation and Opti-Prep density gradient fractionation

2–4 confluent 150 mm dishes (BD, San Jose, California) of cells grown in 2% FBS to limit excessive amounts of growth factors, were rinsed with ice-cold PBS and scraped into homogenization buffer (20 mM Tris-HCl, pH 7.8, 250 mM sucrose (Sigma), 1 mM $CaCl_2$ and 1 mM $MgCl_2$). Cells were pelleted by centrifugation at 4°C for 5 min at 250 $g$ and resuspended in 1 ml of cold homogenization buffer supplemented with protease and phosphatase inhibitors. Cells were then subjected to mechanical disruption with 15 strokes of a tight pestle in a tissue grinder (Kimble/Kontes, Rockwood, Tennessee). Homogenates were then passed through a 22 g needle 20 times. Lysates were then centrifuged at 500 $g$ for 10 min at 4°C to get rid of intact cells, nuclei and cell debris. The resulting supernatant was centrifuged at 52,000 $g$ for 1 hr at 4°C. Supernatants were stored (soluble fraction) and the pellets were subjected to 1% Triton X100, 50 mM Tris-HCl pH 7.4, 150 mM NaCl, 5 mM EDTA supplemented with protease and phosphatase inhibitors. After 20 min incubation on ice, lysates were centrifuged at 52,000 $g$ for 1 hr at 4°C. Supernatants were collected as the triton-sensitive fraction. Remaining membrane pellets were further extracted with 100 mM N-octyl glucoside (Santa Cruz), 50 mM Tris HCl pH 7.4, 150 mM NaCl supplemented with protease and phosphatase inhibitors. Following 20 min of incubation on ice, samples were centrifuged at 16,000 $g$ for 30 min at 4°C. Supernatants were collected as the detergent resistant membrane (DRM) fraction.

For the detergent-free Opti-Prep (Sigma) density gradient fractionation, 4 confluent 150 mm dishes of cells grown in 2% FBS were rinsed and scraped into the homogenization buffer. Cells were pelleted by centrifugation at 4°C for 5 min at 250 $g$ and resuspended in 600 µl of cold homogenization buffer supplemented with protease and phosphatase inhibitors. Cells were then subjected to mechanical disruption with 15 strokes of a tight pestle in a tissue grinder (Kimble/Kontes). Homogenates were then passed through a 22 g needle 20 times. At 4°C, lysates were centrifuged at 1000 $g$ for 10 min. The resulting postnuclear supernatant was transferred into a separate tube. The pellet was again lysed by the addition of 400 µl cold homogenization buffer supplemented with protease and phosphatase inhibitors followed by sheering 20 times through a 22 g needle. After centrifugation at 1000 $g$ for 10 min at 4°C, the second postnuclear supernatant was combined with the first. 1 ml of 50% Opti-Prep (Sigma) was added to the combined postnuclear supernatants and placed in the bottom of a 5 ml centrifuge tube (Beckman Coulter, Brea, California). 400 µl each of 20%, 17.5%, 15%, 12.5%, 10%, 7.5% and 5% Opti-Prep solutions were poured on top of the lysates. Discontinuous gradients were then centrifuged for 90 min at 100000 $g$ at 4°C using an SW-55 rotor in a Beckman ultracentrifuge. After centrifugation, 400 µl fractions were collected and distribution of proteins was analyzed by immunobloting.

## Immunofluorescence

Cells grown on coverslips were fixed with 4% formaldehyde (Polysciences, Inc, Warrington, Pennsylvania) in PBS at RT for 15 min. Coverslips were washed in PBS 3 times for 5 min and blocked in blocking buffer (1xPBS, 5% normal goat serum (Cell Signaling), 0.3% Triton X-100) for 1 hr. Cells were incubated with primary antibodies diluted in 1xPBS, 1% BSA, 0.3% Triton X-100 overnight at 4°C. Followed by rinsing 3x 5 min with PBS, coverslips were incubated with secondary antibodies diluted in 1xPBS, 1% BSA, 0.3% Triton X-100 for 1–2 hr at RT in dark. Cells were then mounted on glass slides using ProLong Gold with DAPI (Cell Signaling) mounting medium and imaged with a Nikon TE2000-U epifluorescence microscope with appropriate filters. Corrected total

cell membrane fluorescence was determined for p-Akt T308 and p-Akt S473 signals using Image J (National Institute of Health, Bethesda, Maryland. Rabbit anti–p-Akt T308, anti-p-Akt S473 and mouse anti-HA were all from Cell Signaling. Alexa 488 anti–rabbit, Alexa 594 anti–mouse from Invitrogen. Quantifications are average of 3 independent experiments of at least 10 cells per experimental condition.

### Cellular proliferation assays

MEFs were seeded in 6-well tissue culture plates (Corning, Corning, New York) at a density of $1 \times 10^4$ cells per well. Cancer cells were seeded in 12-well plates (Corning) at a density of $1 \times 10^4$ cells per well. At the indicated time points, cells were washed with PBS, fixed in 10% acetic acid (Fisher), 10% ethanol for 15 min and then stained with 0.2% crystal violet (Sigma), 10% ethanol for 20 min, washed with distilled water and air-dried. Cell-associated dye was extracted with 2 ml of 10% acetic acid for 15 min and the optical density (OD) was measured at 595 nm. Values were normalized to the OD at day 0 or to vehicle treatment at day seven for each sample. Each data point was determined in either duplicates or triplicates. For experiments performed with limited supplementation, DMEM with 2% FBS was used. For experiments with cancer cells, RPMI with 4% FBS was used. Data shown are the mean of at least three independent experiments with standard deviation.

### Activation of simvastatin

Simvastatin (Sigma) needs to be activated by opening of the lactone ring before use in cell culture. Briefly, four milligrams of simvastatin were dissolved in 0.1 ml of 100% ethanol, with subsequent addition of 0.15 ml of 0.1 N NaOH. The solution was heated at 50°C for 2 hr in a water bath and then neutralized with HCl to pH 7.2. The resulting solution was brought to a final volume with sterile PBS (10 mM), and aliquots were stored at −80°C until use.

### Wound healing assays

Cells were seeded at $5 \times 10^5$ cells/well in 6-well dishes in DMEM, 8% FBS, incubated for 24 hr then serum-starved for 10 hr. A wound was induced into the cell monolayer using a 200 µl pipette tip. Cells were incubated in DMEM, 2% FBS, at 37°C. In triplicate experiments, wounds were imaged at the indicated time points for 12 hr at 3 positions along each wound. Recovery percentages indicate the percentage ratio of distance traversed by the leading edge of migrating cells to the initial wound, determined with ImageJ.

### Transwell migration assays

Cells were seeded in duplicates onto 8-µM PET Transwell migration chambers (Corning) at a density of $1 \times 10^5$ cells/well in media containing 1% FBS and media containing 4% FBS + 20 µM LPA + 5 µM S1P were placed in the lower chamber as chemoattractant. Cells were allowed to migrate to the lower side of the inserts for 18 hr. Cells which did not migrate, were removed with a cotton swab on the upper side and then the inserts were fixed in 10% acetic acid, 10% ethanol for 15 min and then stained with 0.2% crystal violet, 10% ethanol for 20 min, washed with distilled water, air-dried and then scanned. Cell-associated dye was then extracted with 2 ml of 10% acetic acid for 15 min and the OD was measured at 595 nm. Data shown are the mean of three independent experiments with standard deviation.

### Statistical analysis

Two-tailed Student's t-test was used for differential comparison between two groups. T-tests and IC50 calculations were performed in GraphPad Prism.

## Acknowledgements

We thank Dr. Jennifer Spangle for critical reading of the manuscript. We thank the Nikon Imaging Center at Harvard Medical School for technical assistance and the use of instruments. This work was supported by National Institutes of Health Grants P01-CA50661 (to TMR), CA30002 (to TMR), P50 CA168504-01A1 (to JJZ and TMR), P50 CA165962-01A1 (to JJZ and TMR), CA172461-01 (to JJZ),

and Stand Up to Cancer Dream Team Translational Research Grant SU2C-AACR-DT0209 (to JJZ and TMR).

## Additional information

### Competing interests
TMR: A consultant/advisory board member at Novartis. The other authors declare that no competing interests exist.

### Funding

| Funder | Grant reference number | Author |
|---|---|---|
| National Institutes of Health | P50 CA168504-01A1 | Jean J Zhao<br>Thomas M Roberts |
| National Institutes of Health | P50 CA165962-01A1 | Jean J Zhao<br>Thomas M Roberts |
| National Institutes of Health | CA172461-01 | Jean J Zhao |
| Stand Up To Cancer | Translational Research Grant SU2C-AACR-DT0209 | Jean J Zhao<br>Thomas M Roberts |
| National Institutes of Health | P01-CA50661 | Thomas M Roberts |
| National Institutes of Health | CA30002 | Thomas M Roberts |

The funders had no role in study design, data collection and interpretation, or the decision to submit the work for publication.

### Author contributions
OC, Conception and design, Acquisition of data, Analysis and interpretation of data, Drafting or revising the article; JN, SX, Drafting or revising the article, Contributed unpublished essential data or reagents; JJZ, Analysis and interpretation of data, Drafting or revising the article, Contributed unpublished essential data or reagents; TMR, Conception and design, Analysis and interpretation of data, Drafting or revising the article

### Author ORCIDs
Thomas M Roberts, http://orcid.org/0000-0001-6453-3955

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
