## [Decision Letter]

Thank you for submitting your article "Rac1-mediated membrane raft localization of PI3K/p110β is required for its activation by GPCRs or PTEN loss" for consideration by *eLife*. Your article has been favorably evaluated by Charles Sawyers as the Senior Editor and two reviewers, one of whom is a member of our Board of Reviewing Editors. The reviewers have opted to remain anonymous.

The reviewers have discussed the reviews with one another and the Reviewing Editor has drafted this decision to help you prepare a revised submission.

Summary:

The reviewers found the study interesting and the manuscript excellent. Its results illuminate a novel two-step mechanism that differentiates the recruitment and activation of the PI3K p110β isoform and relative roles of GPCR and RAC proteins.

Essential revisions:

1) The correlation of PTEN loss with raft-dependent PI3K function employs pairs of PTEN positive and negative tumor cells lines that differ in a number of other parameters. Conclusions regarding the negative regulation of PI3K signaling by PTEN in rafts vs. nonraft membrane regions should be strengthened by employing PTEN silencing in the p110 mutant and various add-back MEF cells.

2) The behavior of p110α when targeted to lipid rafts is interesting, with LPA induced activation being shown to operate via EGFR. It would be helpful to show that LPA is indeed inducing EGFR activation under these conditions (as in Figure 6), by demonstrating increased tyrosine phosphorylation.

---

## [Author Response]

*Essential revisions:*

*1) The correlation of PTEN loss with raft-dependent PI3K function employs pairs of PTEN positive and negative tumor cells lines that differ in a number of other parameters. Conclusions regarding the negative regulation of PI3K signaling by PTEN in rafts vs. nonraft membrane regions should be strengthened by employing PTEN silencing in the p110 mutant and various add-back MEF cells.*

As an alternative approach, more in conjunction with the experiments using PC3 and BT549 cells in the original draft; we have generated Pten re-expression lines using the parental Pten-null PC3 and BT549 cells. Re-expression of Pten lead to a reduction in p-Akt in both PC3 and BT549 cells, whereas expression of a phosphatase-dead version of Pten did not alter levels of p-Akt. Interestingly, restoration of Pten expression rendered PC3 and BT549 cells more resistant to simvastatin treatment. These results complemented the data obtained upon use of oncogenic p110α mutants in simvastatin treated PC3 cells. Both PC3 and BT549 cells could be partially saved from the adverse effects of simvastatin via expression of Pten, which attenuates the activity of Akt that functions downstream of PI3K signaling. These results further corroborated the notion that Pten loss might be critical in lipid raft mediated p110beta activation. You can now find these results in Figure 7—figure supplement 2.

We also have followed the reviewers’ suggestions and have employed PTEN silencing in MEFs. These results were surprising to us. According to our western blot analysis, we have obtained 2 MEF lines with near complete knock-out of PTEN. As expected, a significant increase in p-Akt and p-S6 levels were detected accompanying the loss of PTEN expression (Figure A). We used these PTEN knock-down MEF lines alongside with control cells to study phenotypes associated with PI3K inhibition and simvastatin treatment. Initially we treated these cells with isoform specific inhibitors of class IA PI3Ks; BYL719 (a p110α specific inhibitor), KIN193 (a p110β specific inhibitor), CAL101 (a p110δ specific inhibitor) or GDC001 (a pan-PI3K inhibitor). wt MEF cells are inherently more dependent on p110α mediated signaling. A modest concentration of p110α inhibitor (1μM) significantly inhibited cellular growth of exponentially growing cells. Comparable amounts of p110β or p110δ specific inhibitors do not significantly affect cellular growth (Figure B). Of note, knock-down of PTEN with 2 shRNAs did not alter this prevalent isoform dependence. PTEN silenced MEFs are still p110α dependent for growth and do not display a significant growth reduction upon inhibition of p110β or p110δ isoforms. In addition, all these MEF lines were exquisitely sensitive to pan-PI3K inhibition by GDC. We have also verified this biochemically in a short-term inhibitor treatment assay analyzing p-Akt and p-S6 in the same MEF lines. Treatment with BYL719 for 2 hours almost completely wiped out p-Akt or p-S6 in either control or PTEN silenced MEF lines whereas KIN193 displayed a much reduced potency in downregulating Akt and S6 phosphorylations in control or PTEN knock-down MEFs (Figure D). In line with the assays using isoform-specific inhibitors, treatment of either control or PTEN silenced MEFs with simvastatin lead to a similar level of growth inhibition. In brief, PTEN loss in untransformed MEFs rendered them neither p110β dependent nor more sensitive to interference with raft function. These results were indeed intriguing given the strong correlation between PTEN loss and the dependence on the p110β isoform.

Our data suggest that although PTEN loss may be essential for progression into a more p110β dependent state, it does not seem to be sufficient for this transition. Indeed, in an animal model of prostate tumor formation induced by Pten loss, in only one out of the three mouse prostate lobes ablation of p110β, but not that of p110α, impeded tumorigenesis with a concomitant diminution of Akt phosphorylation (Jia et al. Nature, 2008). The Vanhaesebroeck lab also found tumor formation driven by PTEN loss was not always blocked by ablation of p110β activity in a different GEM model. Clearly, additional parameters seem to be involved in optimal activation of p110β in a PTEN null setting. At the present time we have not incorporated this data into the text – we feel that negative data is not sufficient to make strong inferences. However if the reviewers or editors prefer we are willing to incorporate this data (probably as a point in the Discussion and supplemental figure).

*2) The behavior of p110α when targeted to lipid rafts is interesting, with LPA induced activation being shown to operate via EGFR. It would be helpful to show that LPA is indeed inducing EGFR activation under these conditions (as in Figure 6), by demonstrating increased tyrosine phosphorylation.*

We have performed the experiment suggested by the reviewers. Western blot analysis for Y1068 phosphorylation on EGFR yielded results which were in complete agreement with previous reports describing activatory EGFR phosphorylation in response to LPA. We have now included a p-EGFR panel in Figure 6 displaying corresponding activation of EGFR upon LPA stimulation.